# The overseas background of Chinese returnee energy scientists

Jin Liu[1], Wenjing Lyu[2]*, Jiaxu Shi[1], Wanrong Liu[1]

**1** School of Humanities and Social Sciences, Beijing Institute of Technology, Beijing, China, **2** Sloan School of Management, Massachusetts Institute of Technology, Cambridge, Massachusetts, United States of America

* wjlyu@mit.edu

## Abstract

In an attempt to uncover the international affiliations impacting the Chinese energy sector, this study applies the method of Curriculum Vitae Analysis (CV Analysis) to explore the overseas background of Chinese returnee energy scientists. The investigation focuses on a representative group of scientists hailing from China's distinguished "985" project research universities. From the available online CVs, we gathered data and identified the United States, Japan, and the United Kingdom as the primary host countries that facilitate the growth and learning of these energy scientists. We also noted a concurrent surge in scientists return to China after acquiring academic and professional experience in prestigious global universities. This study thereby illuminates the evolving patterns of Chinese energy scientists' global mobility and return migration.

## Introduction

China has emerged as a formidable competitor in the global science and technology landscape [1], and it is widely argued that the return of overseas Chinese scientists has played a crucial role in China's recent advancements [2], propelling it towards becoming a science and technology superpower [3]. Particularly in the energy field, China has made significant strides in the past few decades. With the aim of achieving carbon neutrality by 2060, China has devoted substantial efforts to reducing carbon emissions and enhancing energy efficiency through research and development in the energy domain. Previous studies have indicated that the rise in research productivity can be attributed to the recruitment of overseas scientists [4].

Academic mobility is a topic of universal significance [5], and universities worldwide place great importance on encouraging scholars to pursue international experiences. Long-term international experiences during the early stages of a researcher's career have a profound and far-reaching impact on their long-term productivity [6]. While numerous studies have examined the international mobility of scientists and scholars [4, 6–8], research on academic mobility within the energy sector is scarce. This scarcity can be attributed to two main factors. Firstly, there is a lack of extensive literature on academic mobility specifically within the energy sector. Secondly, the energy sector encompasses a wide range of industries, making it a challenging area to investigate. The sector's broad scope presents complexities that impede easy exploration.

**Data Availability Statement:** In light of the potentially sensitive individual information contained in the original CV data of the energy scientists, we have decided not to publicly share the original data. However, interested parties can submit a reasonable data request to the data

access committee of this paper (digital@mit.edu) to gain access to the original confidential data. In addition, we have uploaded the aggregated and anonymized dataset that was used to generate all the figures and tables presented in our manuscript. The aggregated anonymized dataset is included as the Supporting information Files.

**Funding:** Authors who received all grants: J. Liu Grant details: General Project of National Natural Science Foundation of China (71774015); General Project of National Natural Science Foundation of China (71974012). NO - Include this sentence at the end of your statement: The funders had no role in study design, data collection and analysis, decision to publish, or preparation of the manuscript.

**Competing interests:** The authors have declared that no competing interests exist.

This paper aims to address these gaps by utilizing China as a case study to explore four key questions: (1) How are talent flows in China's energy sector connected to the global landscape? (2) Which countries/regions are closely associated with academic mobility in China's energy field? (3) Are there any potential shifts in the patterns of international academic mobility over time? (4) What is the current situation regarding the return of energy talents to China?

Within top universities, scholars undertake significant research activities in the energy field. Given that overseas experience can contribute to enhanced academic outcomes, this paper seeks to investigate the international experiential patterns of Chinese returnee energy scientists working in research universities. In this context, the term "energy scientists" refers to researchers engaged in scientific investigations within universities, holding positions of associate professors or professors, and affiliated with work units encompassing energy power, new energy, or clean energy-related keywords. To conduct this study, information on energy scientists was manually collected from the websites of 40 Chinese research universities, followed by thorough data cleaning and analysis of the international experiences of Chinese returnee energy scientists.

The remainder of this article is structured as follows: Section 2 provides a review of the research background and current academic literature pertaining to this subject. Section 3 elucidates the data sources and research methods employed. Section 4 presents a comprehensive overview of the international mobility of Chinese energy scientists, while Section 5 summarizes and discusses the findings. Finally, Section 6 offers pertinent policy recommendations.

## Research background and literature review

### The rapid progress of China's research in the energy field

To gain insights into China's advancements in science and research within the energy field, we initiated our analysis by examining energy-related scientific papers and patents, which serve as significant indicators of innovation. To identify scientific papers in the energy domain, we conducted a search on the Web of Science (WOS), a prominent scientific database. Fig 1 illustrates the top ten countries in terms of energy scientific paper publications from 2000 to 2020. Notably, the number of papers published by Chinese energy scientists has experienced remarkable growth.

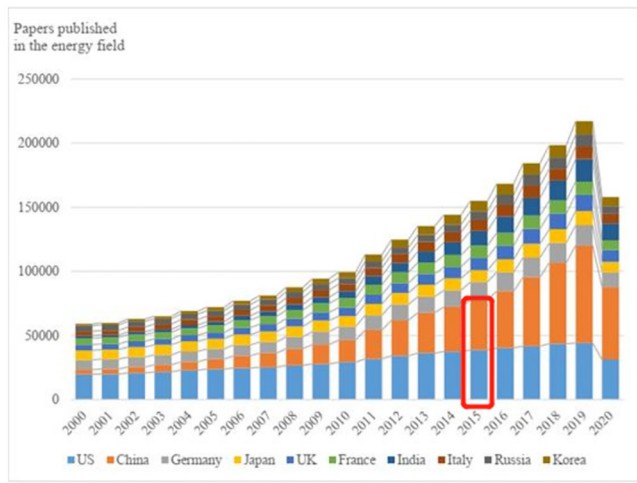

**Fig 1. The top ten countries in terms of publishing scientific papers in the field of energy from 2000 to 2020.**

In 2000, China ranked seventh in terms of energy paper publications. However, by 2005, the number of papers published by Chinese energy scientists had reached a comparable level to that of Germany, Japan, the United Kingdom, and other leading nations. Subsequently, from 2006 onwards, China secured the second position in energy scientific paper publications. Notably, in 2015 (highlighted by a red rectangle in Fig 1), China surpassed the United States, emerging as the top country in terms of total scientific papers published in the energy field. Since then, China has consistently maintained its leading position, even amidst the challenges posed by the COVID-19 pandemic in 2020.

Furthermore, our analysis also focused on the subfield of energy scientific paper publications within the realm of technology and physical sciences. The top ten countries in terms of scientific energy paper publications in these two subfields from 2000 to 2020 are presented in S1 Fig.

In addition to scientific papers, China has also established itself as a formidable force in energy patents. This study utilizes the Derwent patent database to collect information on patentees and generates statistics on the top ten patentees in the energy field. Fig 2 showcases the top ten patentees over the years.

From the perspective of patentees in the energy field, in 2000, no Chinese entities appeared in the top ten list, with Japanese firms occupying eight positions. However, by 2010, renowned Chinese universities such as Zhejiang University and Tsinghua University made their way into the top ten patentee list. The turning point came in 2015 when China boasted five firms/universities among the top ten patentees in the energy field, firmly establishing itself as a superpower in energy inventions. Since then, Chinese firms and universities have consistently maintained a dominant presence in energy patents.

Even in 2020, amidst the challenges posed by the COVID-19 pandemic, nine Chinese firms/universities secured positions in the top ten patentee list in the energy field. Notably, the State Grid Corporation of China, which ranked first in energy patents, accounted for approximately 30% of the total patents filed by the top ten patentees (as shown in Fig 2), with a remarkable 1,665 energy patents issued.

Meanwhile, we also checked the top ten energy patentees in additional patent database: PATSTAT. The results stay consistent. Actually, search results from the PATSTAT suggest that Chinese patentees occupied eight positions in 2015 (compared to five in Derwent patent database), and all top ten energy patentees are Chinese entities in 2020 in the PATSTAT

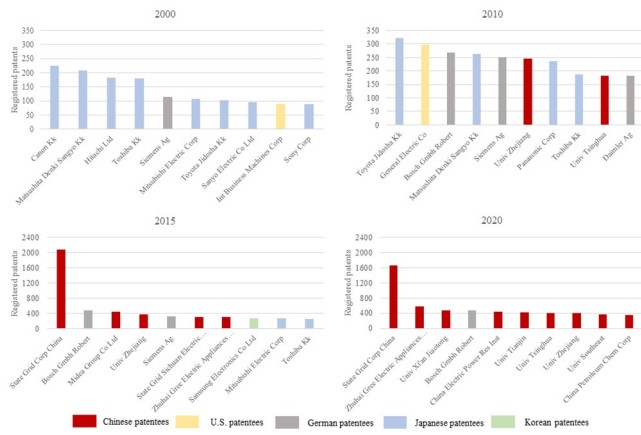

**Fig 2. Top ten patentees in the energy field from 2000 to 2020.**

(compared to nine out of ten in Derwent patent database). The Top ten energy patentees from the PATSTAT database are reported in S2 Fig.

The remarkable growth of China's research power in the energy field primarily gained momentum around 2010, and particularly in 2015. Existing literature emphasizes that the return of overseas talents has played a pivotal role in China's recent advancements in research and development [1]. Notably, returnee scientists tend to publish more papers in prestigious journals compared to their domestic counterparts [1]. This higher productivity can be attributed to their strong connections within the global academic network. Notably, an assessment of China's Young Thousand Talents policy, aimed at attracting top-tier talent, suggests that returnee scientists have made significant contributions to China's scientific achievements [9]. However, prior studies have paid limited attention to the international experiences of Chinese returnee scientists, particularly within the energy field. Consequently, this study aims to bridge this gap by examining the nuanced patterns of international mobility among Chinese energy scientists.

## The important role of energy scientists in China's economic growth

China has set ambitious targets for achieving energy efficiency, which are integral to attaining its broader goals [10, 11]. During his speech at the 75th UN General Assembly, Chinese President Xi Jinping announced China's commitment to reaching peak emissions by 2030 and striving for carbon neutrality by 2060 [12]. However, the outbreak of the COVID-19 pandemic has had an impact on energy development [13]. In response, China has aimed to transform the crisis into an opportunity and guide economic recovery by prioritizing greater productivity. Energy efficiency serves as a primary driving force for reducing energy consumption, while simultaneously supporting China's transition towards a modern, sustainable economy [11, 14].

Research and development efforts in the energy sector are closely linked to enhancing energy efficiency. China's energy conservation and emission reduction initiatives depend on technological advancements and innovations in energy-saving technologies [15]. Inglesi-Lotz, in analyzing the energy research output of five countries based on the International Energy Agency (IEA), highlights those higher levels of economic growth can stimulate greater knowledge capabilities and higher-quality human capital. According to endogenous economic growth theory, improved human capital and increased knowledge accumulation can fuel economic growth across different nations [16]. Numerous recent studies underscore the significance of research and development in the energy sector, emphasizing its role in addressing energy and environmental challenges [17–21]. These studies indicate that China has strategically recruited top scientists and engineers, progressively establishing itself as a major contributor in the field of science and technology. To achieve substantial advancements in energy outcomes, China must prioritize the training and development of energy talents and scholars [22], emphasize their role, and augment the academic output of existing energy researchers.

In contrast to countries like the United Kingdom and the United States, China's research university structure operates in a top-down manner [23]. Consequently, the responsibility for energy policy, scientific research, and technological tasks has primarily fallen on the shoulders of energy scholars within universities. Therefore, this study specifically focuses on universities that undertake significant energy research tasks and examines scientists working in the energy field within these institutions.

The training trajectory of university faculty members plays a crucial role in influencing research outcomes. Hence, it is imperative to conduct a detailed investigation into the training path of energy scientists working in universities. This article specifically concentrates on the

academic training stage of these scientists. Firstly, the current energy faculties in universities generate research outcomes in the energy field, making the training process highly relevant to academic output [24]. Secondly, comprehending the training path of scientists at the present stage holds substantial reference significance for enhancing the training methods for practitioners in the energy field in the future. Furthermore, it provides valuable references and guidance for the future development of the energy industry.

## The importance of overseas experience to Chinese energy scientists

Does transnational education experience lead to higher productivity? The concept of mobility transition was initially proposed by Zelinsky in the 1970s, suggesting that a significant number of international immigrants or skilled personnel and professionals would emerge in the future [25]. While some studies highlight certain drawbacks of academic mobility, such as academic inequality and the concentration of academic centers [26–28], a considerable body of research indicates that international scholars can enhance academic output. Economic studies emphasize the crucial role of transnational mobility in boosting productivity [4, 29]. Furthermore, it has been established that transnational education mobility is linked to individuals' educational trajectories [30]. Pursuing higher education abroad offers individuals a valuable opportunity to gain an international perspective through immersive cross-cultural experiences. Transnational educational experiences facilitate the acquisition of knowledge, attitudes, and skills necessary for working in an international environment [31]. The circulation of highly skilled talent serves as a foundation for the knowledge economy, and international mobility during academic pursuits contributes to institutional internationalization while shaping students' future career prospects and lifestyles [32, 33]. In fact, the mobility of the scientific workforce is a crucial prerequisite for capacity building and achieving global excellence in a particular sector [34].

China did not hold a leading research position in the energy field prior to 2000. To acquire advanced skills, scientists often need to seek education overseas. Highly skilled scientists with transnational educational experiences contribute to the research landscape of traditional energy research countries and bring an international perspective to domestic universities. Therefore, this research focuses on examining the educational experiences of China's energy faculties. By mapping out the current international exchange landscape of China's energy experts' educational backgrounds, this study provides valuable insights into the international mobility experiences of current in-service energy scientists in China. It also offers a reference point for the return of high-level energy talents.

The question of who in Chinese energy faculties engages in transnational mobile education has not been extensively investigated in the context of the energy field in China. While numerous studies have emphasized the significance of transnational mobility, its specific implications for energy scholars in China remain unexplored. As mentioned earlier, countries in the Global North, such as the United States and Europe, are often regarded as traditional science centers and dominate various scientific disciplines [34]. Given this, it is worth examining which countries offer Chinese energy scholars opportunities for transnational communication and exchange in the energy field. Are these countries the same as the destinations for overseas exchanges among Chinese energy scientists? Does overseas exchange experience facilitate their ability to secure positions at prestigious universities? Additionally, it is important to consider whether the choice of overseas exchange destinations and countries differs based on the current universities in China. These questions necessitate empirical data for a comprehensive understanding, and we will address them in Sections 3 and 4 of this study.

## The "brain-circulation" in China's energy field

In the past decade, a new phenomenon known as talent circulation has emerged in the scientific field, involving high-level, highly skilled individuals who study or work in other countries and subsequently return after a short period. This movement is typically driven by the pursuit of scientific research opportunities [35]. Research on the long-term impact of academic mobility in Germany has demonstrated that talent circulation initiated a process of academic mobility and collaborative accumulation, greatly facilitating Germany's reintegration into the international scientific community after World War II [32]. The phenomenon of brain circulation has garnered attention in various industries as well. Several studies have explored the flow of individuals obtaining graduate degrees and the occurrence of talent circulation, with a focus on mathematics and economics [29, 36]. However, there is still a gap in understanding the flow of professional scientists in the energy field.

While brain drain resulting from transnational mobility poses challenges, it is gradually evolving into a talent cycle. Many studies have indicated that countries such as Eastern Europe, Central Europe [37], India, South Korea, China [38–40], and other Asian countries are experiencing this talent cycle. Research on talent circulation in different countries and industries has recently gained momentum. However, there remains a notable dearth of research in the energy field. Investigating the current situation of energy scientists who have pursued overseas study or exchange experiences and subsequently returned to work in their home country can provide insights into stemming the outflow of talent, thereby addressing issues of academic inequality and concentration and promoting the return of high-level talents.

## Methodology

### Research method

The study conducted in this article harnesses the power of Curriculum Vitae Analysis (CV Analysis), a method designed to scrutinize the educational background and career movement trends of scientists/researchers. This comprehensive approach considers not just the conventional higher education pathway, but also the indispensable postdoctoral phase that forms a pivotal part of professional training, especially in the energy field [32, 41, 42].

CV analysis is a potent tool that extracts salient facts from CV data, employing quantitative research techniques to reach insightful conclusions. This approach allows for gathering of both longitudinal and horizontal data pertaining to the research subjects, which assists in mapping out the growth trajectories and mobility patterns of scientists. CV analysis, with its rich history, has found its application in various research contexts [43, 44].

The strength of CV analysis lies in its capacity to vividly depict geographic locations and personnel movement data. It yields sequential data that evolves with time, thereby fusing geographic and temporal information. Its value is particularly evident in studies centered around scientists' background and collaboration. The detailed information offered by CVs can also serves as robust proof of mobility patterns among scientists [45, 46]. In recent years, the CV analysis method has seen a rise in popularity and has been employed across various disciplines such as computer science and economics. In the context of energy research, traditional mobility studies involving professors and researchers are somewhat scarce due to the field's interdisciplinary nature. Energy scientists often collaborate across various universities, departments, and groups, complicating traditional mobility study approaches [47, 48]. However, CV analysis shines in these circumstances with its capability to trace a researcher's full career journey, providing a more precise measurement of geographic mobility [49].

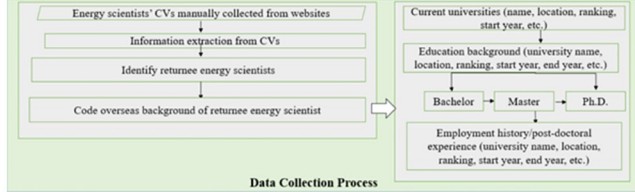

**Fig 3. Data collection process.**

## Data collection process

The procedure for gathering data in this study comprised several stages. Initially, we manually gathered all accessible Curriculum Vitae (CVs) for faculty and scientists specializing in the energy field from 39 Chinese research universities participating in the "985" project. The universities associated with the "985" project is among the most esteemed institutions boasting the most robust research capacities in China. Following this, we meticulously extracted vital data from these CVs, pinpointing energy scientists who have gained educational or professional experience abroad. A detailed breakdown of the data collection process is provided in Fig 3.

The choice of universities involved in the "985" project was guided by the intent to secure a broad and representative sample, taking into account elements such as robust faculty teams and potent research competencies in the energy domain. These selected institutions span across China, ensuring a geographically diversified sample. It's important to note, however, that regional differences in academic strengths might lead to discrepancies. Consequently, there's a denser concentration of universities within East China compared to other regions.

Following the CV collection, we engaged in a meticulous process of manually extracting all vital details. These details encompass the energy scientist's personal information, educational background, and postdoctoral work experience. Table 1 offers a snapshot of the kind of information extracted from the energy scientists' CVs. Any CVs found lacking in information were excluded from the research sample. Moreover, we manually supplemented the extracted data with each university's rank, as per the Quacquarelli Symonds University Ranking (QS Ranking). This globally recognized and widely adopted ranking system is among the most reputable and authoritative of its kind. After eliminating CVs with incomplete data, we were left with a comprehensive set of 1,608 energy scientists' CVs. These CVs contained complete data for all the indicators specified in Table 1 and were retained for further analysis. The amassed CVs furnish a rich data source which will be examined to discern the mobility trends and overseas experiences of Chinese energy scientists.

## Information extracted from the CV

In the data collection process, various information was extracted from the CVs of energy scientists. The following information categories were considered:

**Personal information.** Personal information encompasses essential details about the energy scientists, including their full name, affiliation with educational institutions, and other pertinent information. Collecting personal information is necessary to acquire valuable CVs and ensure representation from various regions of China.

**Educational background.** The educational background forms a fundamental part of the CVs and encompasses all levels of education attained by energy scientists. This section includes key indicators such as the educational level, the names and geographical locations of

**Table 1. Information extracted from energy scientists' CVs.**

| CV information categories | CV information sub-categories | CV Information Index Classification |
|---|---|---|
| Personal information | Current working institution | The name of each energy scientist |
| | | The name and location of each energy scientist's current working institution |
| | | The QS ranking of the current working institution |
| Education background | Bachelor degree | The name and location of the institution where the energy scientist earned his/her bachelor degree |
| | | The QS ranking of the institution |
| | | The enrollment and graduation year of the bachelor degree |
| | Master degree | The name and location of the institution where the energy scientist earned his/her master degree |
| | | The QS ranking of the institution |
| | | The enrollment and graduation year of the master degree |
| | Ph.D. | The name and location of the institution where the energy scientist earned his/her Ph.D. degree |
| | | The QS ranking of the institution |
| | | The enrollment and graduation year of the Ph.D. degree |
| Postdoctoral working experience | Post-doc I | The name and location of the institution where the energy scientist completed his/her first postdoc training |
| | | The QS ranking of the institution |
| | | The starting and ending year of the postdoc training |
| | Post-doc II | The name and location of the institution where the energy scientist completed his/her second postdoc training |
| | | The QS ranking of the institution |
| | | The starting and ending year of the postdoc training |

institutions where the energy scientists obtained their degrees, as well as the years of enrollment and graduation for each educational stage.

**Postdoctoral working experience.** Postdoctoral experience holds significant importance in the career progression of energy researchers. This section of CV information collection entails capturing details such as the names and geographical locations of all postdoctoral positions held by the energy scientists (as exemplified in Table 1 with the first two postdoctoral experiences), along with the starting and ending years for each postdoctoral role.

**QS world university rankings.** The study utilizes the QS university rankings as a standard to measure the level of universities. Three intervals are used to categorize universities based on their QS rankings: top 200, 201–500, and outside the top 500 or not listed in the rankings. This classification helps in analyzing the mobility patterns and experiences of energy scholars across different levels of universities.

By extracting and analyzing this information from the CVs, the study aims to gain insights into the educational and mobility experiences of energy scientists in China and their association with university rankings.

## Results

### The host country/region distribution of Chinese returnee energy scientists

In the analysis of the host country/region distribution of Chinese returnee energy scientists, it is observed that most of the overseas mobility destinations for Chinese energy scientists are developed countries. The results are presented visually in Fig 4, which illustrates the

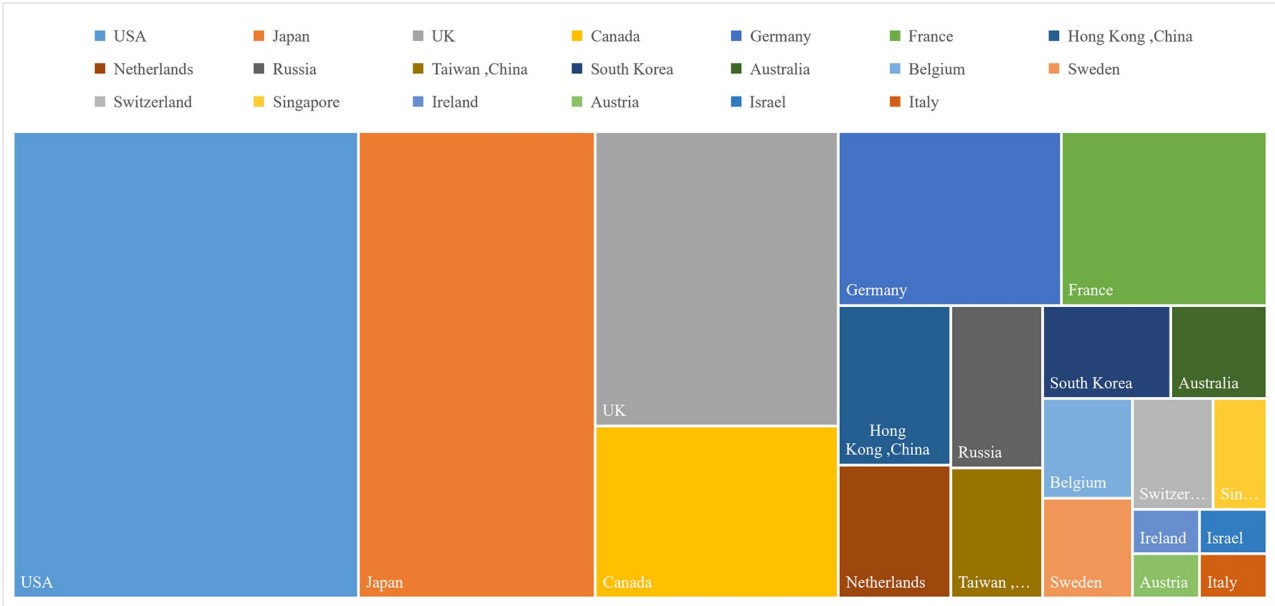

**Fig 4. The host countries/regions distribution of Chinese returnee energy scientists.**

distribution of Chinese returnee energy scientists' international experience across different countries and regions.

The top three host countries for international mobility among Chinese energy scientists are the United States, Japan, and the United Kingdom. Together, these three countries account for 59% of the total sample of returnee energy scientists. Additionally, other notable host countries for Chinese energy scientists' study include Canada, Germany, and France.

To further analyze the trends and proportions of Chinese returnee energy scientists in the top three host countries, Fig 5 is presented. The data show that prior to 2004, Japan was the preferred destination for Chinese energy scholars for overseas exchange, followed closely by the United States, which has traditionally been an academic center in the energy field. However, after 2005, there was a significant drop in the number of Chinese energy scholars

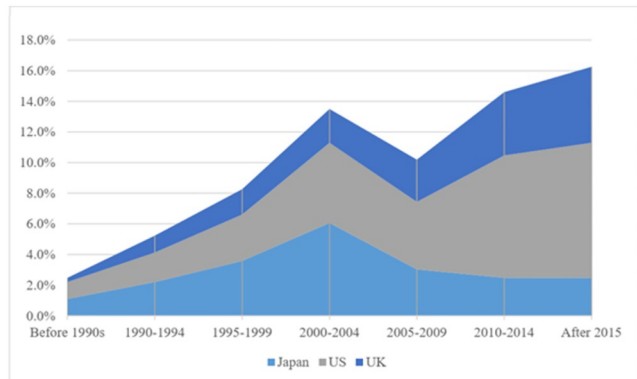

**Fig 5. Trend in the proportion of Chinese energy scientists who have pursued studies in the US, Japan, and the UK.**

studying in Japan, while the United States gained momentum and surpassed Japan. Meanwhile, the United Kingdom's presence as a preferred destination for Chinese energy scholars has been steadily rising since the 1990s and overtook Japan after 2010. This indicates a shift in preference over time, with Chinese energy scholars increasingly favoring the UK over Japan for their studies.

Overall, the results highlight the prominent role of developed countries, particularly the United States, Japan, and the United Kingdom, as major hosts for Chinese energy scientists' international mobility.

## The institution types of Chinese returnee energy scientists' overseas experience

The analysis of the institution types or characteristics of Chinese returnee energy scientists' overseas experience reveals interesting trends. Fig 6 illustrates the changes in the institution types where Chinese energy scientists have studied abroad.

The findings show that over half of the universities where Chinese energy scientists engage in academic exchange overseas are ranked in the top 200 in the QS rankings. Before the year 2000, universities in the top 200 accounted for 53.6% of the institutions where Chinese scientists pursued academic exchange abroad. From 2000 to 2005, this proportion was 50.6%, which increased to 66.7% from 2006 to 2010. After 2010, the figure rose again to 68.9%. This indicates a steady increase over time, with over half of the scientists studying in QS universities ranked in the top 200.

Furthermore, there is a gradual decrease in the proportion of scientists attending universities ranked in the 201–500 range or outside the top 500. These figures stand at 10.8% and 8.1% respectively, suggesting that Chinese energy scientists' choice of universities for overseas exchange is shifting towards higher-ranked institutions.

It is worth noting that the proportion of research institutes and laboratories among the total number of institutions for Chinese energy scientists' overseas exchanges has gradually increased over time. Before the 21st century, this proportion was 4.8%, which has now risen to 12.6%, marking an increase of 2.7 times. This trend indicates that prestigious laboratories, such as the Los Alamos National Laboratory, which attracts top scientists from around the

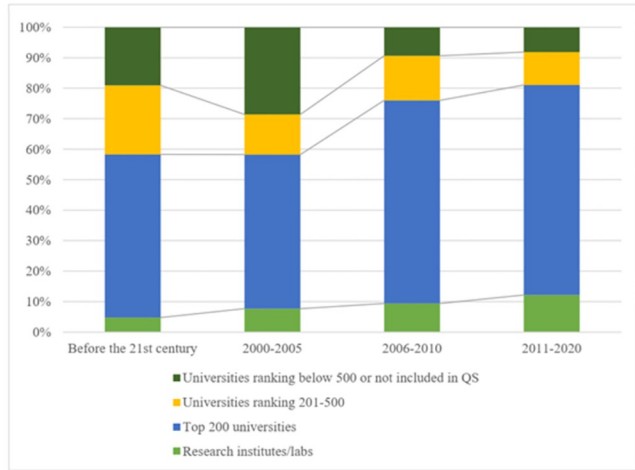

**Fig 6. Evolution of institution types where Chinese returnee energy scientists who have studied or worked.**

world, are included in the exchange institutions. This reflects the level of overseas exchange institutions visited by Chinese scientists and the growing emphasis on the quality of these institutions.

In summary, the analysis highlights a rising trend of Chinese energy scientists opting for higher-ranked universities and research institutes for their overseas exchange experiences. This indicates a shift towards institutions of higher caliber and underscores the importance of prestigious laboratories in attracting Chinese scientists.

## The relationship between the current employment university ranking and overseas experience

The analysis explores the relationship between the current employment university of Chinese energy scientists and their overseas experience. The study focuses on the most recent educational experience, including post-doctorate work, before the scientists' current employment. The data is classified based on the geographical location of the universities or institutions (China or abroad).

The findings reveal that there is a higher proportion of scientists with international education backgrounds in higher-ranked universities. During their undergraduate and master's periods, most Chinese energy scientists did not undertake overseas exchange experiences. The proportion of overseas exchange experience during the doctoral period is also relatively small, accounting for only 11.7%. However, during the postdoctoral period, there is a substantial proportion of scientists who undergo professional training in overseas universities and research institutions. This indicates that overseas mobility experiences during the postdoctoral period contribute to better training opportunities in higher-ranked universities.

Fig 7 presents the distribution of local and returnee energy scientists across universities of different rankings. The findings highlight a correlation between the university's ranking in the QS (Quacquarelli Symonds) ranking system and the proportion of energy scientists with international experience.

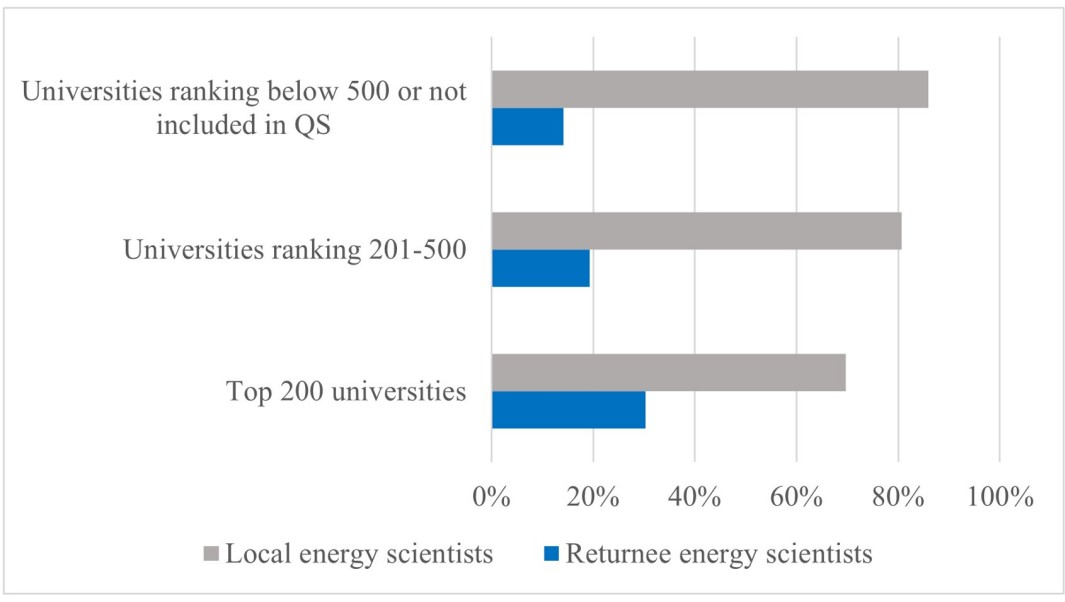

**Fig 7. The distribution of local and returnee energy scientists across universities of different rankings.**

Among energy scientists employed in the top 200 QS-ranked universities in China, more than 30% of them have acquired overseas experience before joining their positions. In contrast, among scientists in Chinese universities ranked between 201 and 500 by QS, less than 20% have gained international experience. For energy scientists working in Chinese universities below the QS 500 ranking, only 14% of them have obtained overseas experience.

These results indicate that higher-ranked universities exhibit a stronger preference for energy scientists with international educational backgrounds compared to less prestigious universities. Additionally, prestigious universities find it easier to attract energy scientists from overseas.

Continuing the analysis, Fig 8 examines the proportion of Chinese returnee energy scientists in different countries/regions based on the levels of universities where they currently work. More than half of the energy scientists in the top 200 QS-ranked universities go to the US, followed by Japan. This suggests that scientists working in the top 200 universities are more inclined to study in the United States. The proportion of scientists working in universities ranked outside the 201–500 and 500 universities (outside the QS rankings) who have studied and exchanged in the US and Japan is also notable. The United Kingdom is also a leading destination for Chinese energy scientists' study and exchange, but the trend differs from the United States. Only 5.13% of scientists in the top 200 universities choose the United Kingdom.

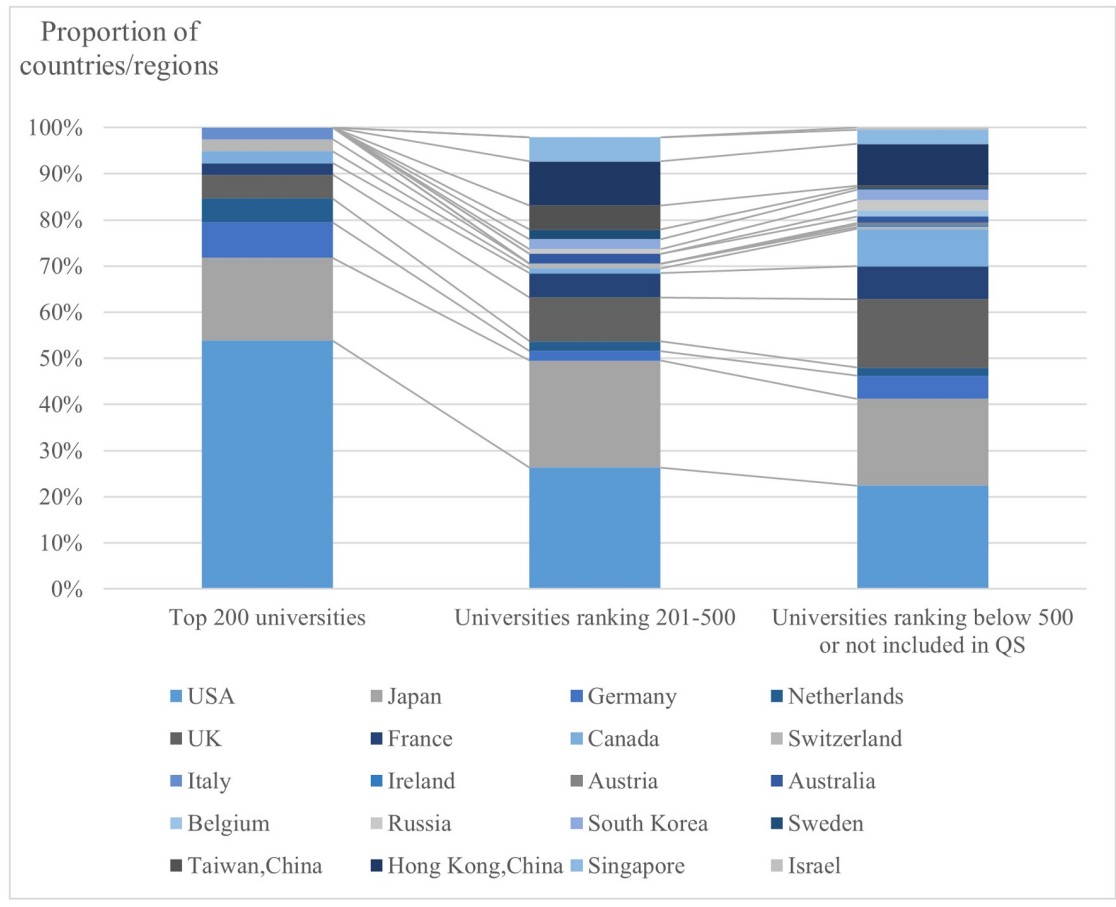

**Fig 8. The proportion of countries/regions from which Chinese returnee energy scientists obtained their education, based on different university rankings.**

However, the proportion of scientists working in QS-ranked 201–500 universities who have studied in the United Kingdom increases to 9.47%, approaching the former. Moreover, 14.80% of scientists working in universities outside the QS top 500 rankings have study and exchange experiences in the United Kingdom.

In summary, the analysis indicates that scientists in higher-ranked universities have a higher proportion of overseas exchange experience. The better the university's ranking, the more significant the requirement for an international education background, attracting scientists with such backgrounds and fostering new academic circles. Additionally, there are varying preferences for countries/regions among scientists based on the levels of universities where they work, with the US being a prominent destination for scientists in top-ranked universities, while the United Kingdom shows a significant presence among scientists in lower-ranked institutions.

## The overall academic mobility pattern of Chinese returnee energy scientists

The analysis of the overall academic mobility pattern of Chinese returnee energy scientists is presented through a Sankey diagram in Fig 9. The Sankey diagram illustrates the mobility across undergraduate to master's, master's to Ph.D., postdoctoral experiences, and current work organizations.

The following conclusions can be drawn from the analysis:

1. There are limited overseas exchange experiences in the academic education of Chinese energy scientists, and the doctoral stage is the primary stage for studying abroad. Chinese energy scientists tend to engage in exchanges during their doctoral and postdoctoral periods. However, there is still a lack of strong connections and exchanges with international standards, indicating that many universities in China remain relatively closed in terms of

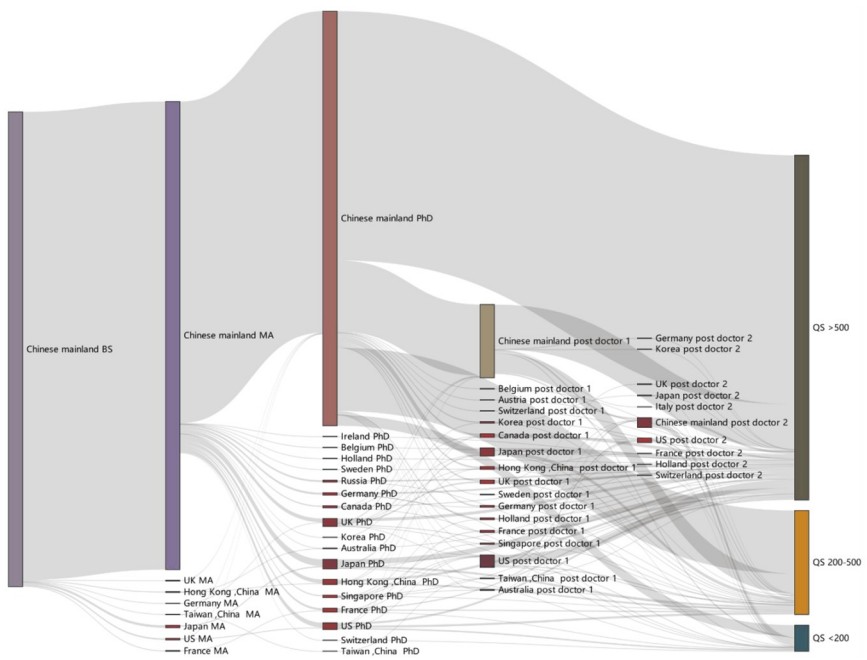

**Fig 9. The Sankey diagram of overall academic mobility of Chinese energy scientists.**

academic engagement. The proportion of students going abroad to study increases as the level of education progresses, with the doctoral stage having the highest proportion of individuals undertaking overseas exchanges.

2. The United States, the United Kingdom, and Japan remain the top destinations for overseas exchanges in the energy field for Chinese scientists. These countries have made significant contributions to the development and training of Chinese energy scholars. Scientists from China's top universities show a preference for studying in the United States, with many attending prestigious universities such as Harvard, Cornell, and Stanford. During the doctoral stage, there is a diverse distribution of destination universities, with 30 institutions from the United States alone, accounting for a quarter of the institutions.

3. During the postdoctoral period, the proportion of exchanges with other countries and regions increases significantly. The postdoctoral stage remains a crucial period for conducting exchanges with various countries worldwide. The proportion of Chinese energy scholars with overseas exchange experience notably increases during the postdoctoral period. Among scholars with postdoctoral experience, 37% have been in overseas universities or institutions. For those with a second postdoctoral experience, the proportion rises to 45%. Additionally, among those with a first postdoctoral experience in mainland China and a second postdoctoral experience, 53% chose overseas institutions. This indicates that many energy scholars with postdoctoral experience prefer to undergo training in overseas universities or institutions to gain an international perspective and enhance their professional capabilities.

4. The proportion of high-level universities has gradually increased, indicating that China's efforts to attract high-level scientific research talents from overseas are yielding results. The distribution of returnees working in the energy field across various universities shows a high degree of polarization. World-class universities like Tsinghua University and Peking University already have an excellent talent structure, with approximately one-third of their scientists having overseas exchange experience. However, the situation is different in other Chinese universities, where the number of scientists with an international education background is relatively small, accounting for less than one-fifth of the total.

In summary, the analysis highlights the patterns of academic mobility among Chinese returnee energy scientists, emphasizing the preference for overseas exchanges during the doctoral and postdoctoral stages, the leading destinations of the United States, the United Kingdom, and Japan, and the increasing proportion of high-level universities benefiting from China's efforts to attract overseas scientific research talents.

## Discussion and implications

The findings of this study have important implications for the development of China's energy sector and the cultivation of scientific research talents. Policymakers and stakeholders should recognize the significant contribution of energy scientists with international education backgrounds and focus on strategies to attract and retain these talents in the Chinese energy sector.

To increase training efforts, it is crucial to encourage energy scientists to engage in academic exchanges at leading energy research centers abroad, allowing them to gain advanced knowledge and experience. This international mobility can contribute to the productive development of the energy industry in China. Additionally, efforts should be made to create an attractive environment for these high-quality talents to return to China, bringing their international perspectives and injecting new vitality into the domestic energy academic field.

It is important to conduct detailed tracking and analysis of the educational and career paths of energy scholars with international education experiences, understanding the factors that influence their decisions to return to China. Similarly, tracking the outflow of talent and analyzing why high-level overseas talents choose to work in foreign countries can provide insights into addressing talent retention challenges.

Given China's ambitious goals for carbon neutrality, there is a need for more innovative talents to achieve these targets. Strengthening technological innovation and talent training will be crucial in driving high-quality development. Policy initiatives, such as sending high-tech talents to study overseas, can contribute to strengthening the vitality of innovation and advancing the energy technology revolution. By introducing high-end talents and promoting the development of green energy as a strategic industry, China can decouple economic development from carbon emissions and improve the efficiency of green energy [50, 51].

However, it is important to recognize that China still faces challenges in carbon reduction, particularly in key industrial sectors with high carbon emissions [52, 53]. Therefore, attracting high-level energy talents to China and leveraging scientific research output in the energy field can play a crucial role in driving emission reduction efforts and achieving sustainable development.

Furthermore, the establishment of a new talent visa system by the Chinese government has facilitated the influx of overseas talents and led to increased academic output in the energy sector [9]. However, several challenges still need to be addressed to bridge the current talent gap in this field. The following aspects should be considered:

Firstly, university policies for attracting energy talents should be targeted and feasible. Targeted policies should focus on addressing the talent gap specifically in the energy sector and implementing strategic supplements accordingly. Feasibility entails incorporating energy scholars into the talent introduction policies of local government departments. Renowned universities like Tsinghua University have actively implemented the "Overseas High-level Talent Introduction Program" since 2009 and have been recognized as innovation and entrepreneurship bases for overseas high-level talents. Their academic reputation and stimulating environment have attracted many scientists with international educational backgrounds, resulting in a robust talent structure. These universities should continue to enhance their appeal and promote academic output. However, universities with lower prestige should adopt more favorable policies and leverage their unique advantages to attract overseas talents. For example, coastal universities can rely on local government welfare and settlement policies, such as Shenzhen's "Peacock Project" and Nanjing's "Nanjing City Talents Housing Measures (Trial)," to attract energy talents.

Secondly, energy scholars should cater to the diverse needs of different talent types and provide flexible conditions for overseas returnees, rather than adopting a generalized approach. The current policy for introducing high-end academic talents encompasses settlement policies, resettlement costs, and provisions for spouses and children. It is crucial to tailor these aspects to the specific needs of different talents. Establishing green channels for high-level overseas talents with unique abilities can maximize their satisfaction and address their individual requirements.

Moreover, while actively attracting the return of energy talents, it is essential to maintain a focus on the input-to-output ratio. A more scientific, transparent, and open evaluation process should be adopted to select talents effectively. Universities should implement talent introduction with a rigorous, detailed, and standardized management system to reform the brain gain system. This approach ensures optimal academic output by using achievements as strict constraints. Additionally, it is necessary to develop a macro-level talent introduction plan that covers talents at various levels, thereby avoiding the pitfall of "blindly introducing talents without

achieving desired outcomes." Furthermore, encouraging academic exchanges overseas after entering universities can gradually establish a domestic-overseas academic ecosystem, which promotes breakthroughs in key energy technologies through high-quality human capital. These efforts will not only contribute to China's goal of achieving peak carbon emissions by 2030 and carbon neutrality by 2060 but also drive progress in the global energy sector and reduce global emissions.

In conclusion, this study highlights the importance of international education experiences for energy scientists and underscores the need to attract and retain these talents in the Chinese energy sector. By focusing on talent cultivation, technological innovation, and international collaboration, China can advance its energy research capabilities and contribute to global efforts in addressing climate change and achieving sustainable development.

## Supporting information

**S1 Fig. Top ten countries in the field of scientific energy papers published in the subfield of technology and physical sciences from 2000 to 2020.**
(TIF)

**S2 Fig. Top ten patent holders in the energy sector from 2000 to 2020 according to the PATSTAT database.**
(TIF)

**S1 Data.**
(XLSX)

## Author Contributions

**Conceptualization:** Jin Liu, Wenjing Lyu.

**Data curation:** Jin Liu, Wenjing Lyu.

**Formal analysis:** Jin Liu, Wenjing Lyu.

**Investigation:** Jiaxu Shi.

**Methodology:** Jiaxu Shi.

**Project administration:** Jiaxu Shi.

**Resources:** Jiaxu Shi.

**Software:** Jiaxu Shi.

**Supervision:** Jin Liu, Wenjing Lyu, Jiaxu Shi.

**Validation:** Jin Liu, Wenjing Lyu, Jiaxu Shi, Wanrong Liu.

**Visualization:** Jin Liu, Wenjing Lyu, Jiaxu Shi, Wanrong Liu.

**Writing – original draft:** Jin Liu, Jiaxu Shi.

**Writing – review & editing:** Jin Liu, Wenjing Lyu, Wanrong Liu.

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
