## [Editor Report · Decision Letter 0]

11 Apr 2023

PONE-D-23-08796Are overseas Chinese energy scientists returning to China? A CV analysis of 40 Chinese research universitiesPLOS ONE

Dear Dr. Lyu,

Thank you for submitting your manuscript to PLOS ONE. After careful consideration, we feel that it has merit but does not fully meet PLOS ONE’s publication criteria as it currently stands. Therefore, we invite you to submit a revised version of the manuscript that addresses the points raised during the review process.

We look forward to receiving your revised manuscript.

Kind regards,

Julian D. Cortes

Academic Editor

PLOS ONE

Journal Requirements:

Additional Editor Comments:

See comments below

Dear author/s, thanks for submitting your work to PLoS ONE ,

I contrasted the core sections of your work with our seven criteria for publication and I consider you could work on the following points before sending it for review:

1 Experiments, statistics, and other analyses are performed to a high technical standard and are described in sufficient detail

Consider expand the description of both bibliographic databases of scientific (WoS) and patent publications (Derwent). Concerning WoS, expand the differentiation between disciplines, areas of research, and subject categories used (i.e., there are four subjects in energy science and technology, which were examined?).

Concerning patents, expand the patent search query and inclusion compare to other patent databases such as PATENTSCOPE or PATSTAT, does this refinement change your outlook?

Consider the accuracy of introducing the ‘big data CV’ term/concept. Since the sample is around ~1600 scientists, it is not clear why this sample should be considered as ‘big data.’

Consider expand the accuracy and completeness of institutional CV for researchers (e.g., out of ~1,600 resumes, how complete the information was retrieved? There was complete information for all variables (module, information, index, indicators) for all scientists?

2 The article is presented in an intelligible fashion and is written in standard English

The article needs copy editing. There are missing words to make coherent sentences (e.g., lines 62-63: “This paper will [missing] China as an example […]; lines 70-71: “ has a total of 2,078 articles, of which [missing] universities author 1,178”). The meaning of core ideas may be misleading given the current state of the paper.

There are sections in bold type for no particular reason (e.g., lines 131-132: “China has set ambitious goals, and improving energy efficiency is the key to achieving these goals”)

3 The article adheres to appropriate reporting guidelines and community standards for data availability

Data Availability

Authors must follow standards and practice for data deposition in publicly available resources including those created for gene sequences, microarray expression, structural studies, and similar kinds of data. Failure to comply with community standards may result in rejection.

The authors checked in the submission that “all data are fully available without restriction.” However, I could not identify any attached file or permanent link.

The authors might consider providing open access to their dataset after anonymizing any personal information and make available in a permanent link (e.g., DOI) in a public data repository. This is extremely valuable for replication/triangulation studies. Here more information on PLoS ONE data availability policy: https://journals.plos.org/plosone/s/data-availability

I hope you can incorporate the above suggestions to improve your already valuable work before sending it to review.

Sincerely,

Julián D. Cortés

Associate Editor

---

## [Author Response · Author response to Decision Letter 0]

15 Jul 2023

July 5, 2023

Dear editors of PLOS ONE,

We appreciate your valuable feedback, and we have carefully revised the manuscript based on your suggestions. Below are our detailed responses to each suggestion:

1.“Please ensure that your manuscript meets PLOS ONE's style requirements, including those for file naming.”

Our reply: Thanks! Our manuscript format has been modified according to PLOS ONE style requirements, including use the correct file names.

2.“We note that you have indicated that data from this study are available upon request. PLOS only allows data to be available upon request if there are legal or ethical restrictions on sharing data publicly. For more information on unacceptable data access restrictions, please see http://journals.plos.org/plosone/s/data-availability#loc-unacceptable-data-access-restrictions. 

b) If there are no restrictions, please upload the minimal anonymized data set necessary to replicate your study findings as either Supporting Information files or to a stable, public repository and provide us with the relevant URLs, DOIs, or accession numbers. For a list of acceptable repositories, please see http://journals.plos.org/plosone/s/data-availability#loc-recommended-repositories.”

Our reply: Thanks! In our revised cover letter, we explained that “In light of the potentially sensitive individual information contained in the original CV data of the energy scientists, we have decided not to publicly share the original data. However, interested parties can submit a reasonable data request to the data access committee of this paper (digital@mit.edu) gain access to the original confidential data. In addition, we have uploaded the aggregated and anonymized dataset that was used to generate all the figures and tables presented in our manuscript. The aggregated anonymized dataset is included as the Supporting Information Files.” 

3.“Consider expand the description of both bibliographic databases of scientific (WoS) and patent publications (Derwent). Concerning WoS, expand the differentiation between disciplines, areas of research, and subject categories used (i.e., there are four subjects in energy science and technology, which were examined?).”

Our reply: Thanks for point out this concern. We also focused on the subfield of energy scientific paper publications within the realm of technology and physical sciences. The top ten countries in terms of scientific energy paper publications in these two subfields from 2000 to 2020 are presented in S1 Fig.

4.“Concerning patents, expand the patent search query and inclusion compare to other patent databases such as PATENTSCOPE or PATSTAT, does this refinement change your outlook?”

Our reply: Thanks for point out this concern. we also checked the top ten energy patentees in additional patent database: PATSTAT. The results stay consistent. Actually, search results from the PATSTAT suggest that Chinese patentees occupied eight positions in 2015 (compared to five in Derwent patent database), and all top ten energy patentees are Chinese entities in 2020 in the PATSTAT (compared to nine out of ten in Derwent patent database). The Top ten energy patentees from the PATSTAT database are reported in S2 Fig.

5.“Consider the accuracy of introducing the ‘big data CV’ term/concept. Since the sample is around ~1600 scientists, it is not clear why this sample should be considered as ‘big data.’”

Our reply: Thanks for point out this. The approximately 1600 energy scientists included in our study give a comprehensive representation of major research universities in China. We have taken note of your feedback and have removed any references or presentations related to "big data" from the manuscript.

6.“Consider expand the accuracy and completeness of institutional CV for researchers (e.g., out of ~1,600 resumes, how complete the information was retrieved? There was complete information for all variables (module, information, index, indicators) for all scientists?”

Our reply: Thanks for point out this. We revised the manuscript to explain the data collection process more clearly. “Following the CV collection, we engaged in a meticulous process of manually extracting all vital details. These details encompass the energy scientist's personal information, educational background, and postdoctoral work experience. Table 1 offers a snapshot of the kind of information extracted from the energy scientists' CVs. Any CVs found lacking in information were excluded from the research sample. Moreover, we manually supplemented the extracted data with each university's rank, as per the Quacquarelli Symonds University Ranking (QS Ranking). After eliminating CVs with incomplete data, we were left with a comprehensive set of 1,608 CVs energy scientists’ CVs. These CVs contained complete data for all the indicators specified in Table 1 and were retained for further analysis.”

So, for the 1,608 CVs we used to conduct all of our analysis, the information for all variables is complete. 

7.“The article is presented in an intelligible fashion and is written in standard English. The article needs copy editing. There are missing words to make coherent sentences (e.g., lines 62-63: “This paper will [missing] China as an example […]; lines 70-71: “has a total of 2,078 articles, of which [missing] universities author 1,178”). The meaning of core ideas may be misleading given the current state of the paper.

There are sections in bold type for no particular reason (e.g., lines 131-132: “China has set ambitious goals, and improving energy efficiency is the key to achieving these goals”)”

Our reply: Thanks for point out this. We have diligently revised the manuscript, ensuring a coherent flow of information, removing unnecessary formatting such as bold displays, and enhancing the language for improved clarity and readability. 

Sincerely,

Wenjing Lyu, Research Associate of Sloan School of Management, 

Massachusetts Institute of Technology, Massachusetts, United States of America.

E-mail: wjlyu@mit.edu

---

## [Decision Letter · Decision Letter 1]

21 Aug 2023

The overseas background of Chinese returnee energy scientists

PONE-D-23-08796R1

Dear Dr. Lyu,

We’re pleased to inform you that your manuscript has been judged scientifically suitable for publication and will be formally accepted for publication once it meets all outstanding technical requirements.

Kind regards,

Radoslaw Wolniak, full professor

Academic Editor

PLOS ONE

Additional Editor Comments (optional):

Reviewers' comments:

Reviewer's Responses to Questions

**Comments to the Author**

1. If the authors have adequately addressed your comments raised in a previous round of review and you feel that this manuscript is now acceptable for publication, you may indicate that here to bypass the “Comments to the Author” section, enter your conflict of interest statement in the “Confidential to Editor” section, and submit your "Accept" recommendation.

Reviewer #1: All comments have been addressed

Reviewer #2: All comments have been addressed

2. Is the manuscript technically sound, and do the data support the conclusions?

Reviewer #1: Yes

Reviewer #2: Yes

3. Has the statistical analysis been performed appropriately and rigorously? 

Reviewer #1: I Don't Know

Reviewer #2: Yes

4. Have the authors made all data underlying the findings in their manuscript fully available?

Reviewer #1: Yes

Reviewer #2: Yes

5. Is the manuscript presented in an intelligible fashion and written in standard English?

Reviewer #1: Yes

Reviewer #2: Yes

6. Review Comments to the Author

Reviewer #1: Dear Authors

I accepted the paper. The Chinese economy is intrusive to other economies of the world, it has changed so quickly. In addition, every reader of this journal will want to know as much as possible about the changes in China.I have one more remark, change the font to black on Figures.

Best wishes

Reviewer

Reviewer #2: The authors made significant corrections to the manuscript. I recommend publication in its present form.

7. PLOS authors have the option to publish the peer review history of their article (what does this mean?). If published, this will include your full peer review and any attached files.

Reviewer #1: No

Reviewer #2: No

---

## [Editor Report · Acceptance letter]

23 Aug 2023

PONE-D-23-08796R1 

The overseas background of Chinese returnee energy scientists 

Dear Dr. Lyu:

I'm pleased to inform you that your manuscript has been deemed suitable for publication in PLOS ONE. Congratulations! Your manuscript is now with our production department. 

Kind regards, 

on behalf of

Professor Radoslaw Wolniak 

Academic Editor

PLOS ONE